# Recipe Based Anomaly Detection with Adaptable Learning: Implications on Sustainable Smart Manufacturing

**DOI:** 10.3390/s25051457

**Published:** 2025-02-27

**Authors:** Junhee Lee, Jaeseok Jang, Qing Tang, Hail Jung

**Affiliations:** 1Data Science Group, INTERX, Ulsan 44542, Republic of Korea; junhee.lee@interxlab.com (J.L.); jay.jang@interxlab.com (J.J.); tangqing@interxlab.com (Q.T.); 2Department of Business Administration, Seoul National University of Science and Technology, Seoul 01811, Republic of Korea

**Keywords:** industry 4.0, injection molding, artificial intelligence, data-driven AI, recipe-based learning, adaptable learning, flexible manufacturing, process optimization

## Abstract

The advent of Industry 4.0 has significantly transformed the manufacturing sector, bringing advancements in quality control efficiency, environmental sustainability, and production development. These changes have led to the development of intelligent technologies such as artificial intelligence (AI). However, implementing AI solutions in manufacturing processes still presents challenges in many aspects, particularly in handling irregular datasets influenced by diverse manufacturing settings. In the field of injection molding, quality inspection often occurs at the batch level rather than at the individual level, providing only the overall defect ratio of batch production instead of labeling each individual product. These issues limit the general application of AI and data-driven decision-making. To address these limitations and enhance product efficiency, this study proposes a novel anomaly detection framework for a specific manufacturing process. In Recipe-Based Learning, we first apply K-Means clustering to account for the flexible manufacturing process, which relies on diverse settings. The injection molding data are classified into setting-specific recipes to ensure data normality and uniqueness. The Kruskal-Wallis test is conducted to provide statistical evidence of differences in data based on varying settings, further justifying the necessity of Recipe-Based Learning. Then, Autoencoders for anomaly detection are trained with normal data from each recipe. With this data-driven AI approach, 61 defective products are predicted, compared to the existing 41 defects. Meanwhile, the integrated model, which does not consider variations in settings, only predicted 2 defects, indicating poor and distorted quality inspection. For Adaptable Learning, which focuses on new inputs with unseen settings, we apply KL-Divergence to identify the closest trained recipe data and its corresponding model. This approach outperformed both the integrated and additionally trained models in predictive power. As a result, continuous prediction is achieved without the need for further training, successfully enhancing process optimization. In the context of smart factories in the injection molding industry, such improvements in process management can significantly enhance overall productivity and decision-making, primarily through a data-driven AI approach.

## 1. Introduction

### 1.1. Background

Industrialization plays a critical role in addressing key challenges related to maintaining viability by introducing cutting-edge technologies in smart manufacturing. Recently, many industries have been accelerating the adoption of Industry 4.0 to enhance quality control and customization efficiency, expecting benefits such as increased production speed and energy efficiency, which can potentially reduce manufacturing costs and environmental impact. However, significant challenges remain in effectively integrating intelligent technologies due to financial, regulatory, and organizational management constraints. Despite these challenges, the future impact of Industry 4.0 on the manufacturing ecosystem is inevitable, driving active research and experimentation into the potential for manufacturing innovation [1,2].

In particular, the integration of new technologies, such as artificial intelligence (AI) in Industry 4.0 has emerged as a key driver of innovation in manufacturing processes. The primary objective of applying AI is to enhance productivity and flexibility across the entire process. Techniques such as machine learning, deep learning, and reinforcement learning contribute to the advancement of data-driven decision-making [3]. Since each manufacturing process has unique characteristics in product development, it is essential to thoroughly analyze the nature of the data. Based on this customized approach, applied algorithms should then be optimized accordingly. Moreover, research findings indicate that the development of AI in the manufacturing sector can help address environmental challenges [4,5]. In other words, as manufacturing AI continues to evolve, it is expected to contribute efficiently to society in various ways.

Many studies have shown that the application of manufacturing AI can drive industry advancements, making it a field that should continue to be developed in the future [6,7,8]. However, despite technological advancements in both hardware and software, the adoption of AI solutions in the manufacturing sector still faces several limitations, such as scalability. Additionally, extensive research is still required to overcome these challenges [9].

### 1.2. Issue

In the field of injection molding, research datasets often fail to sufficiently reflect the realistic characteristics of the entire product development process. These datasets are typically designed with a balanced ratio of normal and defective products to primarily enhance the predictive performance of applied algorithms. Consequently, labeling data on a one-to-one product basis is considered a key element in AI applications [10]. However, in real industrial settings, defective data are significantly scarce, making it challenging to develop AI models that adequately account for this imbalance [11]. In particular, the injection molding process poses additional difficulties in real-time quality inspection for every individual product. Instead, products are generally collected in batches, and quality evaluation is conducted based on the count of defective products within each batch. This approach makes it difficult to determine whether each data point corresponds to a normal or defective product. In cases where defective data are extremely limited, the application of AI for quality inspection becomes even more constrained. Additionally, the injection molding process exhibits distinct characteristics depending on the type of facility and product. This variability suggests that directly applying general AI methods is less effective, necessitating a more specialized and differentiated approach.

Additionally, in the field of injection molding, frequent adjustments to setting values during the process complicate the preparation of training data [12]. When manufacturing settings change, the data itself loses its normality, and the more frequently these changes occur, the more complex the data becomes. Even for a single product, the data may consist of mixed distributions, each exhibiting different characteristics depending on the specific manufacturing settings. If analysis is conducted without considering these diverse settings, there is a high risk of producing distorted results. This issue further complicates the application of AI and hinders effective data-driven decision-making.

### 1.3. Our Idea

This paper proposes a novel framework for applying artificial intelligence (AI) specialized in the injection molding process, primarily using a data-driven approach. Given the characteristics of data with minimal defects, our goal is to develop anomaly detection models that learn only from data of good products in a batch process and subsequently identify actual defects in the remaining data. Additionally, the data distributions vary significantly due to frequent changes in setting values. This highlights the fact that injection molding operates as a flexible manufacturing process, which must be carefully considered in the analysis. To address these characteristics, we introduce the “Recipe-Based Learning” approach. A “Recipe” is defined as a unique set of setting values manipulated throughout the process. If even a single value changes within a set of multiple settings, it is considered a new recipe. The Recipe-Based models are then individually trained for quality inspection. This approach effectively optimizes data for detecting defective products, laying the foundation for data-driven decision-making in process optimization. Furthermore, we introduce the “Adaptable Learning” approach.

Here, we aim to continuously predict new data from unseen settings by leveraging trained models from the closest recipe. Since additional training for new data becomes unnecessary when settings are matched, this approach implies a reduction in computational costs when implementing AI in the manufacturing process. With these advancements, and in the context of smart factories [13], sustainable manufacturing—such as improving energy efficiency and reducing waste—is also expected. For our experiment, we utilized an injection molding dataset with labeling and setting challenges, collected from a private company in South Korea.

### 1.4. Contributions


Optimization in Defect Detection: This paper advances the field of injection molding by employing data-driven AI modeling to improve the accuracy and efficiency of defect detection. The proposed approach is tailored to a specific manufacturing process, enhancing production quality and process optimization.Reduced Training Time: Since additional training for new data is unnecessary, the required learning time is reduced, thereby extending the maintenance period of the product life cycle.Securing Corporate Competitiveness: By lowering the cost of quality inspection through advanced defect detection, companies can strengthen their competitive advantage.Enhancement of Sustainable Manufacturing: When modeling is performed for each individual setting, optimizing model parameters during training becomes simpler due to more distinguishable data characteristics. This not only streamlines the training process but also has the potential to reduce additional computational costs when handling new data, ultimately enhancing overall process efficiency. Furthermore, this approach supports sustainable manufacturing by promoting resource efficiency across the sector. 


This paper is structured to provide a clear progression from foundational concepts to experimental validation and future implications. Section 2 establishes the context for our approach by reviewing the evolution of injection molding technology and recent advancements in AI, emphasizing how our method builds upon and enhances existing techniques. Section 3 details the algorithms and methodologies employed to improve defect detection accuracy, demonstrating their impact on production efficiency. In Section 4, the application of experimental data validates our method’s real-world effectiveness, significantly outperforming traditional approaches in defect detection accuracy. Section 5 interprets these results, highlighting the potential of data-driven AI-based defect detection to enhance sustainability and resource efficiency in manufacturing. Finally, Section 6 advocates for the targeted use of AI in manufacturing automation and proposes future research directions that align with sustainable and resource-efficient practices.

## 2. Literature Review

The injection molding industry has made great strides over time in leveraging AI.

### 2.1. Initial Experimentation and Predictive Modeling (Early 2000s)

Early AI implementations in injection molding focused on predictive maintenance and process optimization, often relying on basic machine learning algorithms to forecast equipment failures and adjust parameters based on historical data. However, these systems faced significant data limitations, as molding processes are highly complex and sensitive to subtle fluctuations in conditions. As a result, prediction accuracy was often inconsistent and required operator expertise to validate predictions, restricting scalability in complex production environments [14].

### 2.2. Integration with Industry 4.0 and Real-Time Monitoring (Mid-2010s)

With the advent of Industry 4.0, AI’s role in injection molding expanded to include real-time data monitoring and adaptive process controls. By integrating IoT sensors and advanced machine learning models, manufacturers could dynamically adjust molding parameters, significantly enhancing efficiency and reducing defect rates. However, challenges remained, particularly in ensuring data compatibility across different machine systems and achieving ‘zero-defect’ manufacturing due to high variability in material properties and environmental factors [15].

### 2.3. Advanced Quality Prediction and Zero-Defect Ambitions (2020s)

More recent advancements focus on achieving high-quality, zero-defect production through sophisticated AI models, such as neural networks and human-in-the-loop systems. These technologies enable AI to refine parameters in real-time, reducing waste and ensuring consistency. However, they require substantial computational resources and skilled data scientists to manage and interpret results, creating a high entry barrier for smaller manufacturers. Additionally, the effectiveness of these systems is often constrained by variability in data quality and completeness, limiting their universal applicability [10].

Over time, AI has become increasingly important in the injection molding industry, with continuous improvements in accuracy. However, frequent changes in setting values make it challenging to effectively utilize AI [12]. Additionally, deploying AI for defect detection and quality prediction in injection molding is hindered by the extremely limited availability of defect data. This scarcity of defect samples restricts AI models’ ability to generalize and accurately predict rare defects [16].

### 2.4. Similar Studies

In fact, in batch-based injection molding processes, Autoencoders have been utilized for monitoring and quality prediction [12,17]. However, practical applications still face significant challenges, particularly due to accuracy degradation caused by changes in new recipes (setting values) [12].

### 2.5. This Work

As explained above, due to the nature of the injection molding process, individual quality labeling is not available for each data point; instead, only the overall defect ratio within cavity-unit-based production can be determined. Therefore, as a data-driven AI approach for defect identification in a specific manufacturing environment, we employed the existing Autoencoder method, which detects defects by learning exclusively from data of good products within the cavity or batch production unit [12,17].

In this paper, we propose a method to secure data normality for more sophisticated modeling and a methodology for accurately predicting new types of data. First, we introduce a learning approach that separates setting values to reflect the flexibility of the overall manufacturing process, as discussed earlier. By classifying setting values, the regularity of the training dataset can be maintained, as illustrated in Figure 1. When modeling is performed based on this regularity, the accuracy and reliability of quality inspection can be significantly improved. This approach enables the effective application of AI for process optimization by fully accounting for the inherent characteristics of the dataset.

Additionally, to efficiently verify new types of input data, we propose a method for making predictions by utilizing the trained model that best matches the new data from the existing Recipe-Based training unit. This is achieved using an algorithm that identifies the distribution of the new data.

## 3. Methodology

In this section, we introduce our proposed architecture for implementing anomaly detection, focusing on datasets that lack individual labeling results and primarily consist of setting values. The general architecture is illustrated in Figure 2. To provide an overview, we first describe the collected dataset used for experimental analysis. Second, we introduce the K-Means clustering algorithm [18], which is applied to secure data normality by separating data based on setting values. Third, we define the train and test datasets for anomaly detection, where only normal data are used for training, and the remaining data are utilized for prediction. Before the preprocessing and modeling procedures, the Kruskal-Wallis test [19] is conducted on the training data to statistically validate differences in input distributions across various setting groups. Data regularization is then performed using min-max scaling before proceeding with the modeling process. Fourth, predictive modeling is conducted using an Autoencoder [20] model, where results exceeding the optimized reconstruction error threshold are classified as anomalies. Finally, for adaptable learning, KL-Divergence [21] is employed to identify the closest matching data among trained recipes when compared to new, unseen settings. The trained Autoencoder model from the closest matching recipe is then used to directly predict new data.

### 3.1. Data Collection

The injection molding dataset used in this experiment is based on a batch measurement procedure, as illustrated in Figure 3. The dataset is primarily collected from multiple facilities using sensor-based techniques. The data gathered via sensors are then merged, enabling comprehensive data analysis and the application of AI methods. Each product consists of either a single-cavity or multi-cavity mold. In each process cycle, a single-cavity mold produces one product, while a multi-cavity mold produces multiple sub-products. For example, if a four-cavity mold produces three normal sub-products and one defective sub-product, the defect ratio for that product is 25%. These cavity-based products are then assigned to a specific batch. In this process, quality inspection is typically not performed on each individual product; instead, the overall defect ratio is determined based on the count of good and defective cavities. Since individual labeling is absent, only the overall defect percentage is available, making it impossible to track specific defect types or their severity. As a result, defect detection is limited to estimating the number of defective cavities collectively. This constraint makes it challenging to generally apply AI-based classification or anomaly detection methods. Additionally, the lack of individual defect labels prevents the measurement of conventional accuracy scores. Furthermore, if setting values change multiple times throughout the process, even a dataset with a single item code may consist of multiple distributions. In such cases, domain-specific data preprocessing is required to ensure reliable analysis and modeling.

The experiment dataset is described as follows.
The collected dataset has a size of ***D*** with ***N*** features.The dataset comprises ***F*** facilities and ***P*** products.Each product consists of ***C*** cavities.***S*** setting features are used to classify data into recipes.***I*** input features are used for training Autoencoder models.

Additional features include metadata such as facility and product codes, process dates, and other relevant information. The target feature for prediction is the overall defect ratio, determined by the count of good and defective cavities in the batch process.

### 3.2. Recipe Separation

The primary objective of this study is to accurately reflect data characteristics based on changes in settings, or in other words, the flexibility of the manufacturing process. The primary objective of this study is to accurately reflect data characteristics based on changes in settings, or in other words, the flexibility of the manufacturing process. We then compare the prediction results of trained models using setting-based classified data—ensuring data normality—with those trained on the original(integrated), unclassified dataset. Through this approach, our work aims to enhance the prediction accuracy of defective cavities, focusing on a data-driven AI approach.

To classify data into recipes based on specific settings, the K-Means clustering algorithm [18] is applied to the original product dataset.

K-Means clustering is an unsupervised algorithm used to group data with similar characteristics based on a distance measure. The process follows these steps:First, the number of clusters K is determined to specify how many clusters the data should be divided into.Initial centroids are assigned for each cluster. Data points are then allocated to the nearest centroid using distance-based metrics such as Euclidean distance.The centroids are updated by moving them to the center of their assigned data points. This process is repeated iteratively until all data points are assigned to clusters and the centroids stabilize.

An example of the K-Means clustering result is shown in Figure 4.

Next, The customized process of applying K-Means is as follows.
From the dataset of size ***D***, select only the ***N*** setting features.Identify ***C*** unique combinations of setting features from the selected data.Set the number of clusters parameter to ***K***, matching the number of unique setting combinations ***C***.Apply the Standard-Scaling method to normalize different scales of setting features.Train the K-Means Clustering algorithm using the final dataset of ***d*** rows and ***N*** featuresMap the trained cluster assignments back to the original dataset of size ***D***, predicting each data point’s cluster based on its setting features.Assign a cluster number from 1 to ***K*** for each data point, defining setting-specific subsets.Define the cluster number of each subset as a setting parameter-based recipes.When new data are collected, predict their cluster (Recipe) numbers using the trained K-Means model. The model assigns one of the existing cluster numbers from 1 to ***K***, determining which recipe the new data should belong to. This ensures the reproducibility in setting-specific fault (defect) detection models.
Figure 4K-Means Clustering Example.
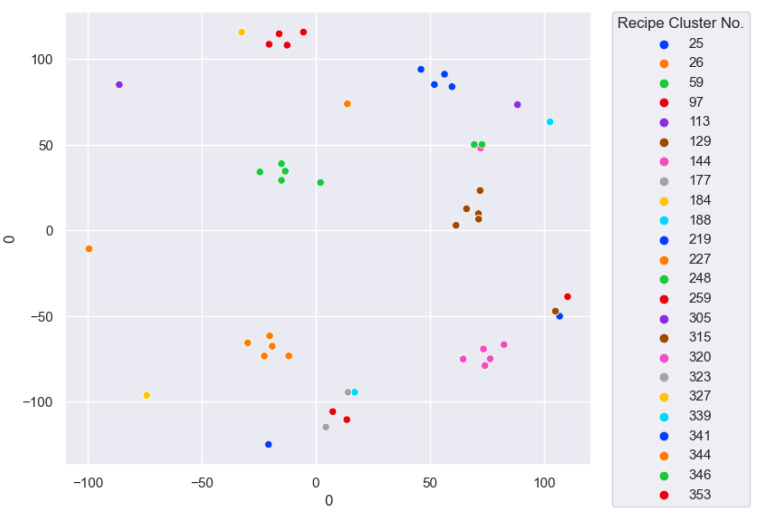



The outline of the K-Means application is shown in Figure 5.

### 3.3. Train/Test Data Organization

As mentioned in Section 3.1, data from this process lack individual labeling results; instead, only the overall defect ratios, based on the count of good and defective cavities, are available. For anomaly detection, batch processes with no defects (normal quality) are first defined as the training dataset. To prevent overfitting during the modeling process, 90% of the dataset is used for training, while 10% is allocated for validation. Finally, both the integrated normal dataset and the classified recipes are used for training the models.

Then, batch processes containing defects (defect ratio > 0.0%) are then defined as the test dataset. To ensure accurate predictions, additional preprocessing is performed in two specific cases.
Suppose a batch process consists of multiple setting-based recipes, some of which also exist in the training data. While this may seem suitable for prediction, a major setback arises: since only the overall defect ratio per batch process is available, comparing the predicted defect ratios of multiple recipes to the actual defect ratio is inappropriate. In other words, within a batch process, the defect ratio for each individual recipe cannot be tracked. Therefore, only batch processes containing a single unique setting recipe are retained in the test data.Suppose the test data contain unseen recipe information. In this case, only partial predictions can be made, leading to an incomplete quality assessment of the batch process.

Finally, the preprocessed test dataset, both in its integrated form and classified by recipes, is used for prediction. By defining both the integrated and Recipe-Based datasets, a comparative predictive analysis can be conducted considering the influence of different settings.

### 3.4. Kruskal-Wallis Test

After organizing the data for the experiment, a statistical test is first conducted on the training dataset (containing only normal data) to provide quantitative evidence that input values vary depending on the changes in settings throughout the manufacturing process. To mitigate the challenge of satisfying strict normality—which can be highly sensitive to slight or severe skewness—the Kruskal-Wallis test [19] is used instead of the one-way ANOVA test [22]. The basic hypothesis for the test is as follows, with a 5% significance level chosen as the standard for statistical hypothesis testing:H0: The medians of all groups are equal, meaning that they originate from the same distribution.H1: At least one group has a different median value, meaning that the groups originate from different distributions.

### 3.5. Min–Max Regularization

Before building prediction models, the Min-Max scaling method is applied to transform the input feature range to [0, 1] [23]. In deep learning, regularization through scaling is essential for effective model training. Especially, it helps mitigate input bias [24], gradient vanishing or exploding [25], covariance shift [26], and ensures exponential convergence of training loss [27]. Scaling is first performed using the training dataset. Then, the same scaling parameters derived from the training data are applied to regularize the features of the validation and test datasets.

### 3.6. Predictive Modeling

#### 3.6.1. Autoencoder Application

To perform anomaly detection using the training and test datasets defined in Section 3.3, the Autoencoder method is employed as the primary prediction model in this study.

The main function of an Autoencoder is to reconstruct inputs into similar outputs using a symmetric architecture [20,28]. The process is as follows:Input data are compressed into a latent vector through the encoder layer.The latent vector captures non-linear correlations between features, effectively learning the most important components of the input data.Using the latent vector, the decoder layer reconstructs the output to match the original input as closely as possible.The ***I*** input features are defined as training data for Autoencoders applied to both Recipe-Based datasets and the integrated normal dataset.

Since the primary objective of this study is to apply AI to a manufacturing process that has been scarcely analyzed in related research, our approach aligns more closely with a data-driven methodology [29]. For this reason, a fundamental Autoencoder structure, as illustrated in Figure 6, is utilized instead of Autoencoder variants [30], which are further modified to enhance model performance—a characteristic of model-driven approaches [31,32].

#### 3.6.2. Threshold Optimization with Evaluation Metrics

After training the Autoencoder models, the reconstruction error between the inputs and predicted outputs is calculated using Mean Absolute Error (MAE). Among various metrics, MAE is chosen for its intuitive representation of the difference between inputs and outputs. The MAE function computes the mean value of absolute errors between the original input values and their reconstructed outputs [33].

The MAE is defined by the following equation:(1)$$\begin{array}{c} MAE=\frac{1}{n}\sum \left|Y_{i}-Y_{i}\right| \end{array}$$
where ***n*** represents the size of the training dataset, $Y_{i}$ denotes the input values, and $Y_{i}$ represents the reconstructed outputs [34].

Furthermore, the thresholds for anomaly detection are determined based on the maximum MAE value of the reconstruction error from the validation normal data. If the MAE value of an input exceeds the threshold, it is classified as anomalous. The application is to be performed in Section 4.5.3.

### 3.7. Adaptable Learning

When new data from untrained recipe settings are collected, additional training is usually required. However, if the model of a trained recipe with the closest distribution to the new settings can be identified, direct prediction becomes possible. This enables the extension of data control beyond the current timestamp. In our work, we define this as Adaptable Learning.

First, KL-Divergence is used to measure the difference between distributions of setting values. The Kullback-Leibler (KL) divergence is a statistical measure that quantifies the distance between two probability distributions. The equation below [21] defines the distributions ***P*** and ***Q*** in our work as follows:***P*** is defined as each setting feature of the trained recipe.***Q*** is defined as each setting feature of the new recipe.The KL-Divergence ranges from zero to infinity. A lower divergence indicates greater similarity between the distributions.Therefore, the goal is to identify ***Q*** that can properly infer ***P***.(2)$$\begin{array}{c} D_{KL}\left(P||Q\right)=\int P\left(x\right)log\frac{P(x)}{Q(x)}dx \end{array}$$

Next, The process of applying KL-Divergence is as follows:Select recipes ***R,…, R + K*** from the trained dataset.Define recipes ***r,…, r + k*** in the new dataset. (Recipe numbers are predicted by the trained K-Means model in Section 3.2).Select ***N*** setting features for each trained recipe and new recipe.Calculate the KL-Divergence value for each setting feature between the new and trained recipes.Calculate the sum of the KL-Divergence values.Identify the trained recipe with the lowest KL-Divergence value compared to the new recipe.Select the model of the trained recipe and use it to predict data for the new recipe.Define ***I*** input features used for prediction.

The outline of KL-Divergence is illustrated in Figure 7.

## 4. Experiment Setup

Before describing the experiments, we were unable to find any public dataset where sensing values fluctuate according to setting values and where defective and good products are labeled based on this distribution. Therefore, we decided to prepare data in an environment where the proposed theory could be practically applied. For this purpose, we used actual injection-related manufacturing process data collected from Company “A” in Cheonan, South Korea for the experiment.

### 4.1. Data Collection

The injection molding dataset used in our work was collected from a private manufacturing company in South Korea and is primarily aimed at producing car steering wheels. The data is first collected from sensors across multiple facilities, followed by a merging process to refine the raw data, making it suitable for AI implementation. As referenced in Section 3.1, the input parameters are defined as follows: ***D*** = 432,089, ***N*** = 132, ***F*** = 4, ***P*** = 14, ***C*** = 1, ***S*** = 76, ***I*** = 6.

For further details, the dataset consists of 432,098 records (data size) and 132 features, representing 14 products manufactured across 4 facilities. Each product is based on a single-cavity system (1 Cavity = 1 Product). A total of 76 setting features are used for recipe numbering, while 6 dynamic input features are utilized for anomaly detection. This structure highlights the high flexibility in setting adjustments throughout the manufacturing process. Table 1 describes the collected process parameters for the setting features. Based on the mixture of settings, the 6 dynamic features (Injection Time, Switch Position, Cushion Distance, Weight Time, Max Injection and Press Peak Pressure) fluctuate throughout the process, potentially causing each feature to follow a multinomial distribution.

### 4.2. Recipe Separation

The dataset is first classified based on unique setting parameters. Then, following the procedure outlined in Section 3.2, the input parameters are defined as follows: ***D*** = 432,089, ***N*** = 76, ***C*** = 366, ***K*** = 366, ***d*** = 366.

An example illustrating the dataset is described in Table 2. In this example, batch number “20231214” represents the date when the manufacturing process took place. The batch process, which follows specific settings defined as recipe number “10”, results in an approximate defect ratio of 1.37 for cavity-based products. As a reminder, the specific types or severity of defects are not specified.

### 4.3. Select Experiment Data

From the complete dataset, we selected the most appropriate subset for applying our proposed method based on the following steps:Remove facility data that do not contain any defects.Select data from a specific facility with the highest defect ratios.Choose data for a specific product with the largest size.

The refined dataset now consists of 26,577 records (data size) with 132 features. It includes 36 setting-based recipes with relatively stable normality, spanning 57 batch processes from 14 December 2023, to 17 July 2024. The descriptive statistics of 6 input features is shown in Table 3.

### 4.4. Train/Test Data Organization

As referenced in Section 3.3, 6 recipes remain in the experimental dataset. The descriptions of the training and test data are provided in Table 4 and Table 5. For recipes 4, 5, and 6, only training is possible since there are no test data with identical recipes. In this case, predictions can only be made if new data for the same recipes are collected. This indicates that, from the original 8190 records, three setting-specific subsets are controllable for a data-driven AI approach. Subsequently, validation data are defined, and Min-Max normalization is applied, as referenced in Section 3.5. With the training and test data fully organized, the prediction process for both the integrated (unclassified by settings) and Recipe-Based models proceeds to the next step in Section 4.5.

### 4.5. Results

#### 4.5.1. Kruskal-Wallis Test

As referenced in Section 3.4, the general hypothesis can be formulated to determine whether input values from the training data vary depending on the recipes (setting-specific groups):H0: The median values of inputs across different recipes are the same, indicating that the inputs originate from the same manufacturing process.H1: At least one recipe has a different median input value, suggesting that the inputs come from different manufacturing process settings.

The *p*-values presented in Table 6 indicate that the 6 input features from the training data exhibit statistically significant differences at the 5% significance level. This leads to the rejection of the null hypothesis, suggesting that input values are not uniform across the entire manufacturing process. Consequently, the data should be trained based on diverse settings rather than an integrated modeling approach.

#### 4.5.2. Autoencoder Configuration

The details of the trained Autoencoders are presented in Table 7. Stacked Autoencoders [35], utilizing the best weights [36] obtained throughout the training epochs, serve as the basic structure. To maximize performance, we optimally selected the parameters [37] for each model.

As mentioned in Section 3.6.1, our work explores the feasibility of applying basic Autoencoders to a specific process using a data-driven approach. Following this logic, rather than performing strict parameter optimization, we focus on an empirical design approach—primarily assessing how well each model captures the distribution of the validation data for normal-quality samples.

First, for the integrated dataset and Recipe 1, which contain more than 4000 samples, the output and inner layers consist of 128 and 64 neurons, respectively, to sufficiently capture the input characteristics. Latent vectors with 16 and 4 neurons are added to effectively handle the complexity of nonlinear inherent patterns through compressed representations. For Recipe 2 and Recipe 3, which contain fewer than 1000 samples, the output and inner layers, along with the latent vector, consist of 64, 32, and 16 neurons, respectively.

Although the largest dataset contains fewer than 10,000 samples, dropout rates between 0.1 and 0.15, along with validation split ratios, are applied to mitigate overfitting during the training process.

With Hyperbolic Tangent as the activation function and the Adam Optimizer—both of which are commonly used for model configuration—all models undergo 200 training epochs. Throughout the training process, models with the best weights are saved for each dataset. The batch size per epoch varies from 10 to 100. To enhance training speed, the learning rate of the Adam optimizer is adjusted from the default 0.001 to 0.01. Additionally, the loss function is defined as MAE, which facilitates an intuitive comparison between input and reconstructed outputs.

#### 4.5.3. Threshold Optimization

To define optimal thresholds for anomaly detection, the validation data referenced in Section 3.3 and Section 4.4 are first predicted using each Autoencoder. The distribution of reconstruction errors is illustrated in Figure 8. The optimal thresholds are determined based on the maximum MAE values, as referenced in Section 3.6.2. Rather than using a statistical approach, the maximum MAE value is selected to ensure the highest level of objectivity. The calculated thresholds are presented in Table 8.

Compared to the training thresholds, the validation thresholds yield smaller values, positioning them relatively closer to the MAE loss distributions. For the integrated data, it is visually apparent that it is prone to false-negative issues, except in cases of extreme values, as both thresholds are positioned too far from the distribution of reconstruction loss. In contrast, for the Recipe-Specific data, the validation thresholds align more closely with the obtained distributions. This suggests that if the reconstruction loss of new inputs closely follows the expected distribution, the likelihood of false negatives is significantly reduced, enabling a more accurate identification of defective data.

#### 4.5.4. Performance Comparison: Integrated vs. Recipe-Based Models

As mentioned in Section 3.1, the dataset lacks individual labeling for defects, preventing a direct comparison of accuracy, false negative rates, and false positive rates using metric scores. Instead, only the counts of normal and defective cavities—without details on defect types or severity—are available based on the overall defect ratio of the process. Consequently, comparisons can only be made by evaluating the count of existing and predicted defects.

With the defined validation thresholds, the prediction results from both the integrated and Recipe-Based models are presented in Table 9. The table includes information on the processed date of each batch product, along with the number of existing and predicted defects. For Recipe-Based predictions, the corresponding recipe number is also provided.

For example, in the first batch product entry, there are 5 existing defects among 971 cavities, with 1 defect predicted by the integrated model and 6 defects predicted by the Recipe-Based model. The recipe number associated with the specific settings of the trained model and test data is 3. Additionally, it can be observed that 8 batch products were generated using the same setting values as recipe number 1.

Compared to the total 41 existing defects in 5842 cavities (0.701%), the integrated model—without accounting for differences in settings—predicted only 2 defects (0.034%), whereas the Recipe-Based models predicted 61 defects (1.04%). In terms of defect prediction, the Recipe-Based approach demonstrated more than twice the accuracy of the integrated model. Notably, in the second and eighth batch productions—both derived from the same setting values (Recipe)—the Recipe-Based model correctly predicted the same number of defects as observed in the actual test data. In contrast, the integrated model failed to predict any defects in these cases.

#### 4.5.5. Statistical Perspective on Data Distribution and Model Performance

As referenced in Section 2.5, the experimental dataset follows a multinomial distribution due to variations in manufacturing settings. From a statistical standpoint [38]—based on the analysis results in Section 4.5.1—this implies that the integrated dataset consists of multiple subsets originating from different distributions, depending on the settings.

When training is conducted without considering these differences, the setting-specific distributions are likely to interfere with one another, introducing noise that disrupts the training process. This interference inevitably leads to distorted results, explaining why the integrated model performs poorly when normality is not preserved for each setting-specific subset in the manufacturing process.

This finding provides clear empirical evidence that when data are separated into setting-specific recipes, normality is better preserved, and differentiation is more effective. As a result, Autoencoders can more accurately distinguish between normal and defect-containing batch productions compared to an integrated model-based approach.

#### 4.5.6. Implications for Model Development and Process Optimization

Furthermore, if thresholds had been determined based on the training data instead of the validation data, the predicted defect ratios would have been significantly lower than the actual defect ratios. This misclassification would have resulted in defective products being labeled as normal, leading to false negative issues. This can be interpreted as a case of overfitting or underfitting caused by training exclusively on normal data.

These findings highlight the importance of designing appropriate prediction models and optimizing thresholds while accounting for variations in data distributions due to changes in settings. In future process optimization efforts, addressing overfitting and underfitting issues will be crucial for improving predictive performance.

### 4.6. Adaptable Learning

Referring to Section 3.2 and Section 3.7, we used new input data predicted as recipe numbers 6 and 7. The characteristics of the new data are as follows:For Recipe 6, there are two unique batches: one with no defects (defect ratio = 0.0%) and another with existing defects (defect ratio > 0.0%). In other words, the first batch consists entirely of good cavities, while the second batch contains both good and defective cavities.For Recipe 7, there is a single unique batch with existing defects.

For both recipes, data containing defects are predicted using the following approaches:Identify the nearest trained recipe data and its corresponding prediction model (Autoencoder) for recipe numbers 6 and 7 using KL-Divergence calculations.Optimize thresholds using validation data, as further described in Section 4.6.2.Predict new data with each selected Autoencoder and the integrated AutoEncoder referenced in Table 7.Compare the prediction results of the integrated and Recipe-Based models.

For the batch process data in Recipe 6 with no existing defects, a new model is trained. This model is then used to predict the batch process data containing defects. This comparison is conducted to evaluate anomaly detection performance between additional training and adaptable learning.

#### 4.6.1. KL-Divergence Calculation

With the defined parameters ***R*** = 1, ***K*** = 2, ***r*** = 6, ***k*** = 1, ***N*** = 76, and ***I*** = 6 as referenced in Section 3.7, the calculation results are presented in Table 10. As a result, the closest trained recipe dataset for the new inputs is identified as Recipe 1.

#### 4.6.2. KL-Divergence-Based Validation Selection

To minimize false positives and avoid overfitting or underfitting issues when predicting the number of defective cavities, the validation dataset is also selected using KL-Divergence calculations. These data are categorized based on whether whether a batch process from the new dataset consists only of normal cavities.

Suppose data from an unseen recipe are collected from two batch productions. The first batch contains no defects, while the second batch includes a certain defect ratio. In this scenario, the first batch is used for validation to optimize new anomaly thresholds based on the trained Autoencoder. The second batch then serves as the new input for the Autoencoder to predict the count of defects.

In contrast, when new data do not contain batch products of only normal quality, the data are directly used for prediction with the trained Autoencoder. In this case, the validation data are defined based on the closest existing recipe that matches the initially selected training dataset.

#### 4.6.3. Train/Validation/Test Data Organization for New Data

The validation data for the new inputs are defined in Table 11.

For Recipe 6, the first batch (Batch No. = 20240902, defect ratio = 0.0%) is used for validation, while the second batch (Batch No. = 20240903, defect ratio > 0.0%) is used for prediction. Second, for Recipe 7, the validation dataset is defined as follows:Based on Recipe 1, select the closest dataset among the remaining Recipes 2 and 3.Define the parameters according to Section 3.7: ***R*** = 2, ***K*** = 1, ***r*** = 1, ***k*** = 0, ***N*** = 76, ***I*** = 6.Using KL-Divergence, identify Recipe 3 as the closest dataset to Recipe 1 as shown in Table 10.Designate the existing training data of Recipe 3 as the validation dataset.

#### 4.6.4. Threshold Optimization for New Data

As referenced in Section 3.6.2 and Section 4.5.3, threshold optimization is performed based on the validation dataset of normal cavities. Using the organized data in Table 11, thresholds for anomaly detection are first optimized based on the maximum MAE values of the reconstruction error distributions. The adaptable learning procedure is then applied to the prediction data.

The definition of thresholds using validation data is illustrated in Figure 9 and Table 12. Initially, the MAE value of the validation threshold from the trained Autoencoder of Recipe 1 is 0.0273, as shown in Figure 8 and Table 8. Since new validation data are now specified, the trained Autoencoder first predicts the data and subsequently updates the maximum threshold value. Due to changes in distributions caused by the closest, yet different settings, the overall values of the reconstruction error distribution naturally increase. Consequently, the maximum threshold values are updated to 0.088 for predicting test data of Recipe 6 and 0.105 for Recipe 7. An exception is observed in the integrated model, where threshold values remain unchanged, as no specific recipe can be identified in this case.

#### 4.6.5. Performance Comparison: Adaptable Learning vs. Integrated & Additional Training Models

The prediction results with updated thresholds, compared to the fixed integrated model, are presented in Table 13. In particular, for Recipe 6, the prediction results of additional training are also included for comparison.

While the integrated model—without considering unique settings—continues to fail in predicting defects, as seen in Section 4.5.4, the adaptable learning process produces relatively acceptable defect predictions. In Recipe 6, predictions from the trained Autoencoder Recipe 1 resulted in significantly more detected defects compared to the existing ones, whereas the integrated model failed to predict any. Notably, adaptable learning demonstrated superior performance over additional training, predicting 10 fewer anomalies and effectively mitigating false positive issues. In Recipe 7, the predicted anomalies using adaptable learning closely matched the actual defects.

#### 4.6.6. Implications for Model Robustness and Process Optimization Through Adaptable Learning

If the thresholds were set to the original MAE value of 0.0273, severe false positive issues would arise due to discrepancies between the actual and predicted defect ratios. This issue stems from overfitting or underfitting. The findings suggest that applying adaptable learning to new data enahances robustness against false positives and false negatives enabling continuous inference. Moreover, adaptable learning demonstrates greater precision in prediction accuracy in defect counting for quality inspection, potentially improving the overall process optimization.

## 5. Discussion

For comparison, both the integrated model and the customized setting-specific recipe models utilized Stacked Autoencoders with a simple structure. Recognizing the importance of hyperparameter tuning [39,40,41], this process was performed for each model.

When applying the trained results of both the integrated model and the Recipe-Classified models, the thresholds of the Recipe-Based models can be set more precisely than those of the integrated model. When used to classify good and defective products, the defective classification rate for each Recipe-Based model significantly outperforms that of the integrated model. This underscores the importance of categorizing controllable data that vary with settings and ensuring the normality of each dataset during training. In other words, it highlights the necessity of training models while maintaining the normality and unique characteristics of data by classifying distributions that vary according to settings, ultimately enhancing production efficiency and process optimization.

Furthermore, when predicting data that have not been previously trained, a KL-Divergence approach to identify the closest recipe distribution demonstrated superior accuracy in distinguishing defective from non-defective products compared to both the integrated model and the additionally trained model. This method expands the control of identifiable settings and reduces the need for frequent retraining while maintaining high classification performance. The targeted nature of Recipe-Specific models streamlines data processing, minimizes noise, and improves accuracy by addressing setting-specific variations that the integrated model may overlook. As a result, these models avoid the broader generalizations required by the integrated model, thereby enhancing defect detection rates and potentially optimizing computational efficiency. This adaptability ensures scalability and robustness, particularly for large-scale industrial applications.

An additional approach leveraging fine-tuned thresholds without extensive retraining further emphasizes the potential of setting-specific modeling. Through continuous inference, this method achieves high prediction accuracy while optimizing computational resources. Such advancements highlight the transformative potential of adaptive modeling in dynamically changing industrial environments, enabling leaner, more efficient AI applications.

In summary, the findings demonstrate the efficacy of AI modeling specialized for a data-driven approach over traditional methods in the injection molding process. These methods effectively address the variability inherent in operational settings, ensuring improved prediction quality and reduced resource demands while adapting to flexible manufacturing processes. The shift toward context-driven methodologies represents a significant advancement in the development of robust and efficient AI systems for sustainable manufacturing.

## 6. Conclusions

As AI becomes integral to manufacturing, its role in fostering efficiency and sustainability is becoming increasingly significant. AI-driven systems can optimize data processing, enabling precise defect identification and waste reduction. This emphasis on quality control not only enhances competitiveness but also can support energy efficiency and minimizes environmental resource waste, contributing to sustainable manufacturing.

When implementing AI in real-world manufacturing sites, its environmental benefits must also be considered. The advantages of AI for environmental sustainability have been clearly demonstrated in small-scale projects and hypothetical or qualitative studies [42,43,44]. However, to achieve a meaningful impact from the perspective of AI and sustainability, it is essential not only to identify and improve environmental factors but also to prioritize those that enhance competitiveness and offer tangible business benefits [45,46].

Compared to existing systems, this approach introduces a new recipe unit-based paradigm for advanced data-driven decision-making, focusing on waste reduction, productivity improvement, model compression, and enhanced prediction accuracy. This, in turn, leads to cost savings in real-time inference. Additionally, it fosters new research directions and promotes continuous development. By leveraging these AI-driven efficiencies, this study highlights that resource waste—an increasingly critical environmental concern—can also be reduced.

The development of optimized, setting-specific datasets enables the deployment of simpler, less resource-intensive models without compromising accuracy. This streamlined data-driven AI approach facilitates continuous inference for quality inspection, potentially reducing computational costs by eliminating unnecessary additional training, thereby fostering broader AI adoption in sustainable manufacturing. Incorporating domain-specific knowledge further refines these models, improving interpretability and predictive capabilities while promoting proactive quality control.

However, in real manufacturing environments, only a limited number of product batches can be accurately labeled as good products, leading to challenges in data quality. To address this issue, semi-supervised learning must also be leveraged to enhance AI-driven approaches. And due to the nature of semi-supervised learning, classification accuracy may not always remain at a consistently high level.

Future research should focus on further integrating domain expertise, advanced feature selection techniques, and optimized data practices based on the Recipe-Level analysis outlined in this study. Specifically, leveraging domain knowledge to identify key variables that influence the distinction between high- and low-quality products, combined with targeted feature engineering, can improve predictive accuracy.

Furthermore, in cases where the lack of specific defect information arises due to limitations in individual quality identification, a more sophisticated approach must be adopted to enhance data-driven process management and generalize the proposed framework effectively. These advancements will ensure that AI systems remain efficient, precise, and sustainable in driving innovation across various industries while effectively addressing critical environmental challenges.

## Figures and Tables

**Figure 1 sensors-25-01457-f001:**
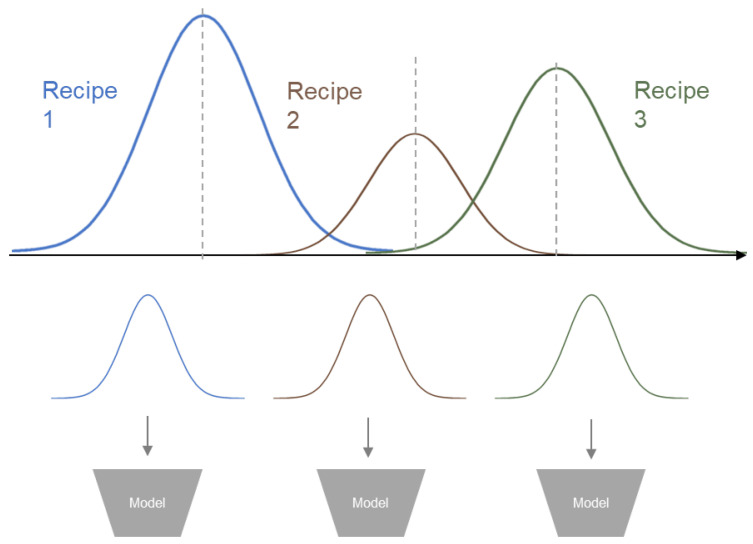
Distribution of Injection Molding data by Settings.

**Figure 2 sensors-25-01457-f002:**
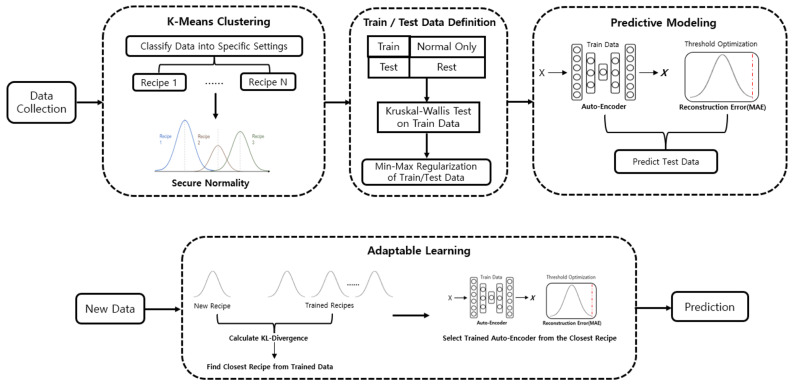
General Architecture of The Proposed Method.

**Figure 3 sensors-25-01457-f003:**
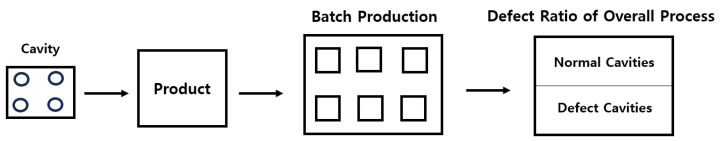
Process Management Based on Batch Production.

**Figure 5 sensors-25-01457-f005:**
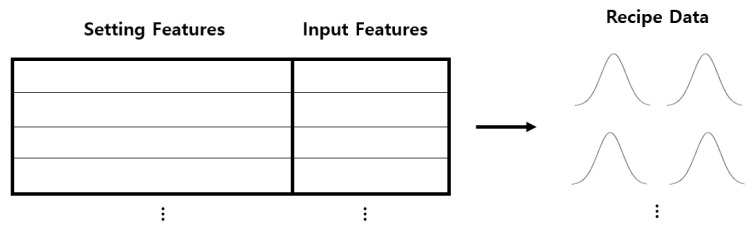
Classify Data into Recipe Settings.

**Figure 6 sensors-25-01457-f006:**
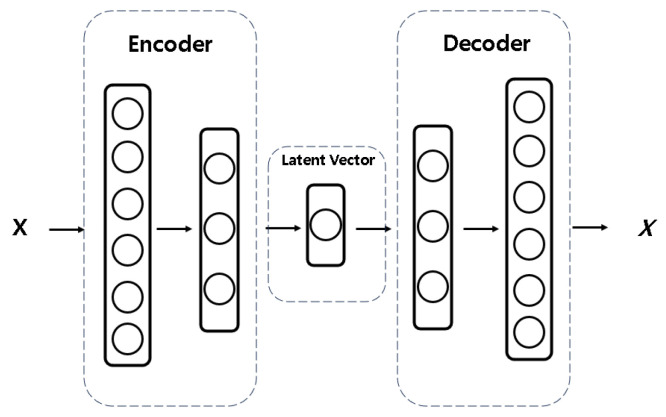
Autoencoder Structure.

**Figure 7 sensors-25-01457-f007:**
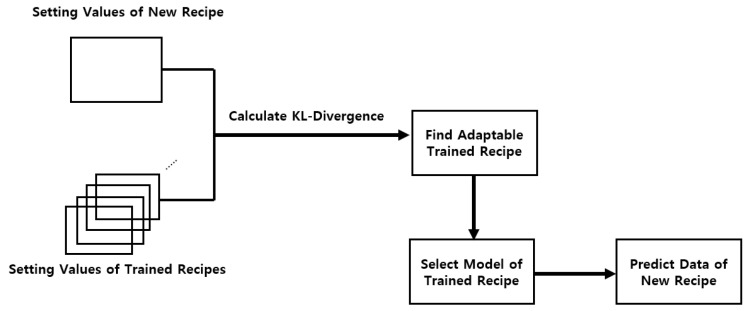
KL-Divergence Application.

**Figure 8 sensors-25-01457-f008:**
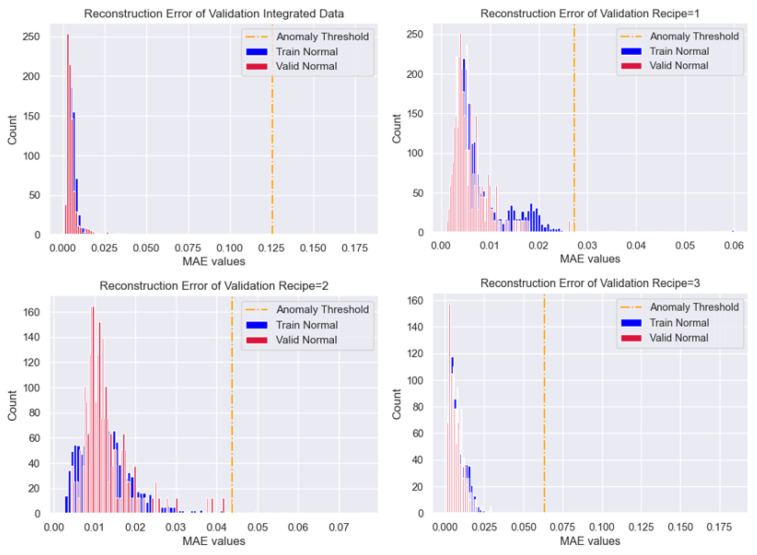
Validation Thresholds of Reconstruction Errors.

**Figure 9 sensors-25-01457-f009:**
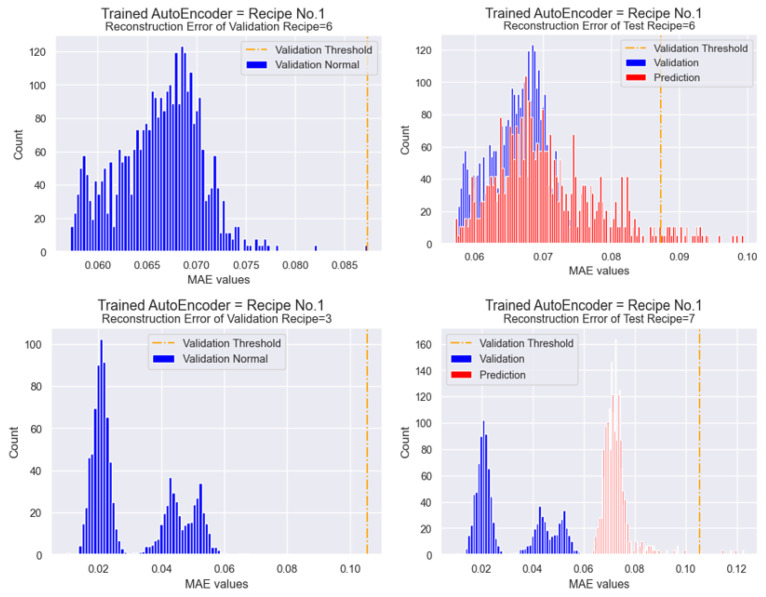
Validation Thresholds of Reconstruction Errors (Adaptable Learning Process).

**Table 1 sensors-25-01457-t001:** Specification of Setting Features.

Quantity	Description
1	Nozzle Temperature
3	H Temperature Condition
5	Weight RPM (Revolutions per Minute)
4	Weight positions
1	Suck Back Position
1	Closing Pressure
3	Closing Speed
3	Closing Position
3	Opening Speed
2	Opening Position
1	Ejection Position
2	Ejection Forward Speed
1	Ejection Forward Pressure
1	Ejection Backward Pressure
1	Ejection Backward Position
1	Ejection Speed
7	Injection Speed
6	Packing Pressure
3	Packing Pressure Time
1	Cooling Time
6	Injection Position
1	Injection Max Pressure
5	Backpress Machine Pressure
1	Short finish Position
1	Hopper Temperature
12	Hot Runners
76	Total Quantities of Setting Features

**Table 2 sensors-25-01457-t002:** Information of Batch process Dataset.

One of the Batch Product Informations of a Specific Recipe	
Batch Number	20231214
Data Shape	438 Data Size and 132 Features
Setting Features	76 Setting Features
Input Features	6 Input Features
Number of Products	438
Predicted Recipe(Cluster) Number	10
Good Cavity Counts(Total)	432
Defect Cavity Counts(Total)	6
Overall Defect Ratio	Approximately 1.37%

**Table 3 sensors-25-01457-t003:** Descriptive Statistics of Input Features.

	Mean	Std	Min	25%	50%	75%	Max
Injection Time	2.33	0.08	2.19	2.30	2.340	2.35	6.39
Switch Position	13.46	2.83	7.99	12.00	13.50	15.99	21.0
Cushion Distance	11.29	2.72	5.88	9.81	11.39	13.69	18.56
Weight Time	24.10	2.76	17.33	23.85	24.26	24.52	172.03
Max Injection Press	151.52	2.52	118.73	149.91	151.40	153.17	171.26
Peak Pressure	13,321.01	12.31	13,260.5	13,312.6	13,323.0	13,330.4	13,358.9

**Table 4 sensors-25-01457-t004:** Train Data Description.

Dataset Information	Data Shape (Existing Defect Ratio = 0.0%)	Test Appliable
Integrated Data	8190 Data Size and 6 Features	O
Recipe 1 Data	2654 Data Size and 6 Features	O
Recipe 2 Data	1996 Data Size and 6 Features	O
Recipe 3 Data	3073 Data Size and 6 Features	O
Recipe 4 Data	435 Data Size and 6 Features	X
Recipe 5 Data	31 Data Size and 6 Features	X
Recipe 6 Data	1 Data Size and 6 Features	X

**Table 5 sensors-25-01457-t005:** Test Data Description.

Dataset Information	Data Shape (Existing Defect Ratio > 0.0%)
Integrated Data	5842 Data Size and 6 Features
Recipe 1 Data	4396 Data Size and 6 Features
Recipe 2 Data	475 Data Size and 6 Features
Recipe 3 Data	971 Data Size and 6 Features

**Table 6 sensors-25-01457-t006:** Kruskal-Wallis Test for Recipes.

Input Features (Training Data)	H-Statistic	*p*-Value (<5% Significance Level)
Injection Time	5299.33	0.0
Switch Position	5579.82	0.0
Cushion Distance	5474.05	0.0
Weight Time	292.93	0.0246
Max Injection Press	1214.03	0.0238
Peak Pressure	1662.22	0.0

**Table 7 sensors-25-01457-t007:** Parameter Descriptions of Trained Autoencoders.

Applied Parameters	Integrated Data	Recipe 1	Recipe 2	Recipe 3
Loss	Mean Absolute Error			
Activation Function	Tanh			
optimizer	Adam			
Learning Rate	0.01			
Number of Epochs	200			
Output Layer Size	128	128	64	64
Inner Layer Size	64	64	32	32
Latent Vector Size	16	4	16	16
Dropout Ratio	0.15	0.1	0.15	0.15
Batch Size	100	10	10	10
Validation Split Ratio	0.15	0.1	0.15	0.15

**Table 8 sensors-25-01457-t008:** Threshold Comparisons for Anomaly Detection.

Autoencoder	Train Maximum Threshold	Validation Maximum Threshold
Integrated	0.180	0.125
Recipe 1	0.059	0.0273
Recipe 2	0.076	0.0437
Recipe 3	0.184	0.0635

**Table 9 sensors-25-01457-t009:** Prediction Comparison of Defect Ratios.

Batch No.	Existing Defect Ratio	Integrated Pred	Recipe Pred	Recipe No.
20240322	0.514 (5/971)	0.102 (1/971)	0.617 (6/971)	3
20240402	0.319 (1/313)	0.00 (0/313)	0.319 (1/313)	1
20240429	0.879 (6/682)	0.00 (0/682)	1.173 (8/682)	1
20240430	0.833 (7/835)	0.00 (0/835)	1.556 (13/835)	1
20240502	0.611 (5/817)	0.00 (0/817)	1.22 (10/817)	1
20240507	1.920 (7/368)	0.00 (0/368)	0.271 (1/368)	1
20240614	1.312 (5/381)	0.262 (1/381)	2.099 (8/381)	1
20240617	0.647 (2/309)	0.00 (0/309)	0.647 (2/309)	1
20240623	0.289 (2/691)	0.00 (0/691)	0.723 (5/691)	1
20240717	0.210 (1/475)	0.00 (0/475)	1.473 (7/475)	2
Total	0.701% (41/5842)	0.034% (2/5842)	1.04% (61/5842)	

**Table 10 sensors-25-01457-t010:** Calculation Result of KL-Divergence Values.

New Recipes	Trained Recipe 1	Trained Recipe 2	Trained Recipe 3
Recipe 6	40.1	41.63	41.12
Recipe 7	11.13	12.66	12.15

**Table 11 sensors-25-01457-t011:** Data Organization for Adaptable Learning.

Existing Training Data	Defined Validation Data	Prediction Data
Recipe 1	Recipe 6 (Normal)	Recipe 6 (Defect)
Recipe 1	Recipe 3	Recipe 7

**Table 12 sensors-25-01457-t012:** Validation Thresholds for Anomaly Detection(Adaptable Learning Process).

Trained Autoencoder	Existing Threshold	New Maximum Threshold
Recipe 1	0.0273	0.088 (Recipe 6)
Recipe 1	0.0273	0.105 (Recipe 3)

**Table 13 sensors-25-01457-t013:** Prediction Comparisons of New Recipes.

Recipe No.	Batch No.	Defect Ratio	Integrated	Adaptable	Additional
6	20240903	0.293 (2/682)	0.00 (0/682)	4.692 (32/682)	6.158 (42/682)
7	20240902	0.546 (4/728)	0.0 (0/728)	0.683 (5/728)	X

## Data Availability

The data are not publicly available due to their containing information that could compromise the privacy of research participants.

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
