# Peer review of "Recipe Based Anomaly Detection with Adaptable Learning: Implications on Sustainable Smart Manufacturing"

_sensors, 2025, doi:10.3390/s25051457_

Round 1

Reviewer 1 Report

Comments and Suggestions for Authors

In the manuscript,the proposed framework integrates K-Means clustering and autoencoders for recipe-based anomaly detection, with KL-Divergence enabling adaptable learning. This addresses the challenge of irregular data distribution caused by frequent parameter changes in injection molding, offering practical value in batch-level quality inspection. Major Revision is recommended to address methodological transparency, experimental validation, and presentation issues. The core idea is valuable but requires stronger empirical support and technical details to meet publication standards.
1. In the section of introduction, further clarification is needed on how this approach differs from existing autoencoder-based methods (e.g., [12,17].
2. The use of K-Means for recipe classification is logical, but the criteria for selecting cluster numbers are not discussed, potentially affecting reproducibility.
3. Autoencoder architecture details (e.g., layers, activation functions, optimizer) are insufficient, so expanding these would enhance technical rigor.
4. KL-Divergence effectively matches new recipes to trained ones, but computational efficiency in real-time industrial settings requires evaluation.
5. Experiments rely on proprietary data, so validation on public datasets would strengthen generalizability.
6. The integrated model’s poor performance needs deeper analysis.
7. Overfitting concerns arise in adaptable learning results, and false positive rates should be addressed.
8. The method’s reduction in training time and improved defect detection is promising, but quantitative evidence for energy efficiency is lacking.
9. Grammatical errors need polishing.

Comments on the Quality of English Language

Grammatical errors need polishing.

Author Response

First of all, we would like to thank you for your plentiful advice. Your comments really have helped us to enhance the logic and complement the flaws of our work. Here are the following responses for your comments.

In the manuscript the proposed framework integrates K-Means clustering and autoencoders for recipe-based anomaly detection, with KL-Divergence enabling adaptable learning. This addresses the challenge of irregular data distribution caused by frequent parameter changes in injection molding, offering practical value in batch-level quality inspection. Major Revision is recommended to address methodological transparency, experimental validation, and presentation issues. The core idea is valuable but requires stronger empirical support and technical details to meet publication standards.

==========================================================

Comment 1. In the section of introduction, further clarification is needed on how this approach differs from existing autoencoder-based methods (e.g., [12,17].

Response 1: 

Thank you for your valuable comment. We appreciate your suggestion regarding the need for further clarification on how our approach differs from existing autoencoder-based methods.

Our autoencoder model follows the same fundamental principles as conventional autoencoder-based approaches. However, rather than focusing on proposing a new autoencoder structure, our study emphasizes the importance of leveraging manufacturing setting values throughout the entire AI application process. This aspect is highlighted in Section 3.6.1 (lines 334–339).

To address your concern and enhance clarity, we have added a brief explanation in Section 2.5 (lines 174–177) to reinforce the objective of our data-driven AI approach.

Thank you again for your insightful feedback.

==========================================================

Comment 2. The use of K-Means for recipe classification is logical, but the criteria for selecting cluster numbers are not discussed, potentially affecting reproducibility.

Response 2: Thank you for your insightful feedback. We understand that your concern primarily pertains to lines 269–275 in Section 3.2, where our explanation may have been insufficient.

To clarify, when determining cluster assignments, particularly for new data, the trained K-Means model predicts the setting-value features of the new data rather than defining new clusters. For example, if the trained model establishes four clusters (1, 2, 3, and 4), any new data point will be classified into one of these predefined clusters, ensuring consistency in cluster assignment rather than generating new clusters.

To address this concern more explicitly, we have revised lines 269–275 in Section 3.2 to provide a clearer explanation. Specifically:

  • We clarified that cluster numbers are determined based on the prediction results, which define the recipe classifications.
  • We explained that when new data is introduced, it is assigned to one of the existing clusters from the trained K-Means model.
  • We briefly mentioned how this approach enhances the reproducibility of setting-specific fault detection models, aligning with the subsequent sections in 3.2.

We sincerely appreciate your valuable feedback and believe that these revisions improve the clarity of our explanation.

==========================================================

Comment 3. Autoencoder architecture details (e.g., layers, activation functions, optimizer) are insufficient, so expanding these would enhance technical rigor. (O)

Response 3 :

 Thank you for your valuable feedback. We acknowledge that the details of the autoencoder architecture were previously insufficient, as only Table 5 presented parameter values.

To enhance technical rigor, we have now included additional details in lines 452–473 in  Section 4.5.2. This revision provides specifications such as the number of layers, activation functions, and the optimizer used.

We appreciate your insightful suggestions and believe this update strengthens the technical clarity of our work.

This version maintains all key points while improving readability and flow. Let me know if you'd like further refinements!

==========================================================

Comment 4. KL-Divergence effectively matches new recipes to trained ones, but computational efficiency in real-time industrial settings requires evaluation.

Response 4 : 

Thank you for your valuable feedback. We acknowledge that our discussion on the advantages of implementing KL-Divergence was presented in an inferential manner, particularly regarding its impact on computational cost. Without quantitative evidence, this may have led to potential confusion.

To enhance clarity, we have revised the relevant sentences—specifically in lines 96–99 (Section 1.3) and lines 679–684 (Section 6)—to explicitly state that the computational efficiency of KL-Divergence is an inferred advantage due to the reduction of unnecessary additional training, rather than a conclusion drawn from direct quantitative evaluation.

We appreciate your thoughtful comments, as they have helped us refine our explanation to better align with the need for empirical validation in real-time industrial applications.

==========================================================

Comment 5. Experiments rely on proprietary data, so validation on public datasets would strengthen generalizability. (O)

Response 5 : Thank you for your valuable suggestion. We fully acknowledge the importance of validating our approach on public datasets to enhance its generalizability.

However, we found that there is a lack of publicly available datasets that accurately correspond to our domain and the manufacturing process-specific dataset used in our study, which was collected from an industrial setting. For this reason, we initially emphasized in lines 105–108 of Section 1.4 that our work is designed to adapt to a specific manufacturing process.

To further clarify this limitation, we have explicitly added this restriction at the very beginning of Section 4(lines 385-390). Unfortunately, due to the absence of suitable public datasets, we leave the broader generalization of our experimental results as an avenue for future research.

We sincerely appreciate your thoughtful feedback and believe that this revision improves the transparency of our study's scope and limitations.

==========================================================

Comment 6. The integrated model’s poor performance needs deeper analysis.

Response 6 : Thank you for your feedback. We see this is very important to emphasize the validity of the first experimental results of our work. Since we have mainly focused on the recipe data , the model trained with integrated data seems to only be shown as a comparison lacking sufficient explanation and quantitative evidence. 

As a review , the primary focus of our study is on improving data-driven methodologies rather than developing new algorithms. Specifically, we emphasize the importance of categorizing data by recipe and ensuring its statistical regularity before training, rather than using a traditional integrated model. Due to this focus, the performance of the integrated model was originally presented primarily as a comparative reference without extensive explanation.

In order to extend the explanation, we first showed quantitative evidence that the input variables differ depending on the settings by conducting the Kruskal-Wallis test on the variables among the different settings, which is added as a new subsection(Section 4.5.1 lines 434-446) before the modeling and prediction procedure from Section 4.5.2 to 4.5.5. For the new subsection, we have also added it to Methodology in Section 3( line199-201, Figure 2 ,new Section 3.4(lines 301-312)). 

By taking this into account, we have added further clarification in prediction comparisons of the Recipe-Based results by segmenting the original draft into  Section 4.5.3, 4.5.4, and 4.5.5. Especially in lines 490-512 from Section 4.5.4 and lines 514–526 from Section 4.5.5, we managed to consecutively explain the reason of the integrated model’s poor performance from the predictive and statistical-based evidence. This revision incorporates key insights, including statistical perspectives, to highlight why training on the entire dataset without considering setting variations can lead to distorted results.

Thankfully , the additional explanations seemed to have led to more compelling logic. We appreciate your insightful comments, which have helped us refine our analysis. 

==========================================================

Comment 7. Overfitting concerns arise in adaptable learning results, and false positive rates should be addressed.

Response 7 : 

Thank you for your valuable feedback. We acknowledge that overfitting concerns in adaptable learning, as well as false positive rates, are important aspects to address.

Before implementing adaptable learning, we ensured that the normal-quality dataset was divided into training and validation datasets, as described in Sections 3.3 and 4.4. Building on this, we also structured the validation data for adaptable learning, which is presented in Table 10 of Section 4.6.2. However, we recognize that our previous mention of overfitting prevention in lines 281–282 from Section 3.3 lacked a comprehensive connection to the overall experimental design.

To provide a clearer and more logically structured explanation, we have made the following revisions:

  • We explicitly stated the rationale for utilizing the validation normal dataset in earlier sections, particularly in the context of threshold designation for defect prediction. This decision was primarily driven by concerns related to false negatives and overfitting (Lines 281–282, Section 3.3).
  • In Table 7 of Section 4.5.3, we incorporated the maximum thresholds derived from the training dataset to facilitate comparison with the validation thresholds and emphasized the importance of threshold optimization (Lines 481–488, Section 4.5.3; Lines 528–536, Section 4.5.6).
  • We enhanced our explanation in Lines 562–574 of Section 4.6.2 and Lines 615–621 of Section 4.6.6, highlighting the importance of updating validation thresholds in adaptable learning, which contributes to the robustness against overfitting and underfitting, ultimately enhancing predictive performance.

Regarding your question on false positive ratios, we acknowledge that our initial explanation primarily focused on false negative issues throughout Sections 4.5.3, 4.5.4, and  4.5.5. However, due to the nature of our dataset, calculating the exact false positive or false negative ratios presents a challenge. So we were only able to mention this issue as a potential problem written generally rather than quantitatively. 

As detailed in Section 3.1, our dataset is derived from a specific manufacturing process where individual defect labels are unavailable. Instead, only the overall defect ratio for cavity-based products across the entire process is recorded. Given these constraints, our approach was to compare the predicted counts of defective cavities as accurately as possible. To improve clarity and prevent potential misunderstandings, we have provided additional details on this limitation throughout the manuscript on Line 219–224 (Section 3.1) and Line 490–495 (Section 4.5.4).

We sincerely appreciate your thoughtful comments, as you have helped us refine our explanations which some were stated just verbally which could have led to fragments of the entire logic and strengthen the logical coherence of our work.

==========================================================

Comment 8. The method’s reduction in training time and improved defect detection is promising, but quantitative evidence for energy efficiency is lacking.

Response 8 : 

Thank you for your insightful comment. As you pointed out, our primary contribution focuses on improving defect detection, while energy efficiency is considered a potential benefit from the broader perspective of sustainable manufacturing. This is the main reason why our study does not provide quantitative evidence specifically for energy efficiency. Additionally, we recognize that this concern aligns with the issue raised in Comment 4.

To enhance the logical coherence of our explanation, we have made the following revisions:

  • We removed the mention of energy efficiency and the comprehensive sustainable manufacturing from the list of keywords in the abstract.
  • We have also briefly mentioned potential contributions to sustainable manufacturing, such as improvements in energy efficiency and reductions in environmental waste, ensuring that these aspects are framed as broader implications rather than primary contributions mainly in the conclusion (Section 6) . 

This revised approach to defect identification and waste reduction through data processing optimization follows a consistent logical structure throughout the manuscript, particularly in:

  • Section 1.4 (lines 115–121): We emphasized our experimental objective, clarifying that the potential benefits extend to sustainable manufacturing rather than being limited to energy efficiency.
  • Section 6 (lines 662–664): We adjusted the phrasing so that aspects of sustainable manufacturing, including energy efficiency, are clearly conveyed as potential benefits rather than direct, quantitatively proven outcomes.

We sincerely appreciate your thoughtful feedback, as it has helped us refine our argument and strengthen the logical consistency of our explanation. Your input was especially valuable in distinguishing between quantitatively supported findings and broader potential impacts for future research.

==========================================================

Comment 9. Grammatical errors need polishing.

Response 9 : 

Thank you for your feedback. 

When taking the above comments into consideration, we found extra grammatical errors and confusing sentences which we haven’t found before. 

We have revised the entire draft for clearer and concise expressions and structures. 

We will make sure there will be no additional errors for the completion of our work and the status of the Sensors Journal. 

Reviewer 2 Report

Comments and Suggestions for Authors

Dear Authors

I have read your paper proposal, and I admit I am confused. I do not know why you submitted your paper to Sensors and there is not a singe measurement or sensor mentioned. Furthermore, you consider your process as a black box. You have not even mentioned what kind of material you inject and what kind of failure do you consider as a defect. Is it dimensional, porosity… As a former quality engineer in the automotive industry (mostly casting and machining of aluminium and ferrous parts) I know the defects have different mechanism. I would expect an explanation why do you consider your paper suitable for Sensors.

Is the defect you are talking about only cosmetical or has some more severe defect.

Is the defect easily detectable or/and are you developing a kind of process control to find the defects.

I do not understand the meaning of the sentence in line 426-428.

Best regards

Author Response

Thank you for your advice. Your comments really have managed to complete our work especially with the concerns of understanding. Thankfully, we have managed to complement the sections that could have been relatively obsurd.

Comment 1. I have read your paper proposal, and I admit I am confused. I do not know why you submitted your paper to Sensors and there is not a single measurement or sensor mentioned. Furthermore, you consider your process as a black box. You have not even mentioned what kind of material you inject and what kind of failure do you consider as a defect. Is it dimensional, porosity… As a former quality engineer in the automotive industry (mostly casting and machining of aluminium and ferrous parts) I know the defects have different mechanism. I would expect an explanation why do you consider your paper suitable for Sensors.

Response 1 : 

Thank you for your insightful feedback. We acknowledge the importance of aligning our paper with the journal's focus and ensuring clarity in our dataset description.

To better correspond with the scope of Sensors, we have added further explanation regarding data collection in lines 210–212 of Section 3.1 and lines 392–395 of Section 4.1, which were previously omitted.

  • The experimental data used in our study was collected from industrial facilities using sensor-based techniques. These sensors were utilized to gather key manufacturing parameters, which were then integrated to facilitate AI implementation and data analysis.
  • The material used in production primarily consists of car steering wheel parts, specifically left and right handle components. We have now explicitly mentioned this information in lines 392–395 of Section 4.1.

Regarding defect identification:

  • Our approach focuses on detecting defective cavities in a batch process. However, instead of having labeled defect data for each individual unit, only the overall quality ratio of the batch is available.
  • Due to missing defect type labels in the dataset we received, we were unable to classify defects based on their specific mechanisms (e.g., dimensional inaccuracies, porosity).

The reason we did not specify the exact material being injected is that our study primarily focuses on applying AI based on data-driven methods to analyze manufacturing process settings and dynamic values rather than investigating material-specific defect mechanisms.

Finally, we submitted our paper to Sensors because our study provides quantitative evidence on how sensor-collected manufacturing data from a specific domain can be effectively leveraged to enhance process efficiency.

We hope this explanation clarifies our rationale and adequately addresses your concerns. Thank you again for your valuable feedback!

==========================================================

Comment 2. Is the defect you are talking about only cosmetical or has some more severe defect.  

Response 2 : 

Thank you for your keen insight. This is a crucial aspect of understanding our experimental dataset, and we appreciate the opportunity to clarify this point.

To address this concern, we have added additional information about the characteristics of our experimental dataset in Section 2.5 (lines 172–177), where we briefly introduce the labeling issue. In short, our experiment does not differentiate defects based on severity, as our approach is based on binary classification.

This decision stems from a fundamental limitation in the dataset we collected. Due to the nature of the manufacturing process, individual cavity-based labeling results were unavailable. Instead, only the overall defect ratio for the entire process was recorded. As a result, our dataset description primarily focuses on the labeling issue rather than distinguishing between different defect severities.

However, considering your comment, we acknowledge that there may be distributional differences among defect types and severities, which would require a more sophisticated AI approach.

To improve clarity and reinforce this point:

  • In lines 219–224 of Section 3.1, we further elaborated on the labeling issue, highlighting the lack of differentiation between defect types and severities.
  • We then connected this discussion to the AI problem outlined in Section 3.1.
  • Additionally, in lines 695–699 of Section 6, we expanded on this issue in the conclusion, recognizing the importance of addressing defect classification in more detail.

We hope our revisions effectively address your concern regarding the lack of specific descriptions of defect severity. Thank you again for your valuable feedback, which has helped us refine our explanations and improve the logical coherence of our work.

==========================================================

Comment 3. Is the defect easily detectable or/and are you developing a kind of process control to find the defects.

Response 3 : 

Thank you for your valuable feedback. This is an important question that helps strengthen the logical foundation of our work.

We utilize an existing methodology in which an Autoencoder is trained on the distribution of previously observed normal (non-defective) data. Since any new data that deviates from this learned distribution is classified as defective, defect detection becomes relatively straightforward when there is a clear distinction between the distributions of normal and defective samples.

However, we identified a limitation in this conventional approach when setting values("recipes") are modified. Specifically, certain variables' distributions deviate from normality under different recipes, which compromises the model’s ability to detect defects effectively. Our work aims to address this issue by improving the robustness of defect detection.

One of the key methodologies proposed in this study is to segment the dataset based on recipes and train the model accordingly. By incorporating recipe-specific distributions, we ensure data normality, optimize the Autoencoder’s training process, and enhance the distinction between normal and defective samples.

Taking your advice into account, we have added defect detection details regarding the recipe-based AI approach in Section 4.5.4(predictive approach) and Section 4.5.5(statistical approach).

We appreciate your insightful comments and hope that our response aligns with your suggestion to reinforce the core logic of our work.

==========================================================

Comment 4. I do not understand the meaning of the sentence in line 426-428.(O)

Response 4 :Thank you in advance for your concern. We acknowledge that some explanations in the manuscript may not have been clearly conveyed, and we appreciate the opportunity to clarify this point.

In particular, we recognize that lines 426–428(Line Numbers from the original draft) provide a definition of the data used in the adaptable learning procedure, which are key components of our approach. However, we agree that the explanation may have been unclear.

To enhance clarity, we have made the following revisions:

  • We added a more detailed explanation in lines 562–574 of Section 4.6.2, introducing two scenarios in batch production to define the validation and test data for adaptable learning.
  • Specifically:
    • If new recipe data includes both normal-only and defect-containing batch products, the normal-only data is used for validation to optimize anomaly thresholds, while the defect-containing data is used for prediction to compare real and predicted defect ratios.
    • If new recipe data includes only defect-containing batch products, it is solely used for prediction. In this case, the validation data is selected based on the closest matching recipe from the training data.
  • Additionally, we further clarified the distinction between validation and prediction data in 576–584 of Section 4.6.3 to reinforce their respective roles.

We hope this revision resolves any confusion and provides a clearer understanding of our adaptable learning procedure. Thank you again for your valuable feedback!

==========================================================

Round 2

Reviewer 1 Report

Comments and Suggestions for Authors

The authors have well addressed all my comments, and the manuscript had been significantly improved; so I recommend the revised manuscript to be accepted for publication.

Author Response

Comments and Suggestions for Authors

The authors have well addressed all my comments, and the manuscript had been significantly improved; so I recommend the revised manuscript to be accepted for publication.

=================================================================================

Response: 

Thank you once again for your keen insights. We have managed to complete the logic for our manuscript with more quantitative evidence and possibly relieve parts that seemed to give confusion. We have checked one more time for any possible mistakes left and managed to upload the final version of our manuscript. 

Reviewer 2 Report

Comments and Suggestions for Authors

Dear Authors

You have considered most of my remarks. I still believe it would be interesting to know which process parameters were acquired. I know there might be some limitations but I believe a statement like 2 pressures 3 temperatures and x should not be a problem, but it will give a reader a better impression about your algorithm and possible implementation.

Author Response

Comments and Suggestions for Authors

Dear Authors

You have considered most of my remarks. I still believe it would be interesting to know which process parameters were acquired. I know there might be some limitations but I believe a statement like 2 pressures 3 temperatures and x should not be a problem, but it will give a reader a better impression about your algorithm and possible implementation.

=================================================================================

 Thank you for your final feedback. As you have mentioned, we have missed some understandings of the dataset collected in real manufacturing sites. And referred to the entire logic of applying AI in a specific manufacturing process, more information of the dataset should have been provided. 

We checked the merged dataset used for experiment and added a new table ( Table1.Specification of Setting Features from Section 4.1.). With the more concrete information of our setting-parameter features, we also added a brief context mentioning that due to the fluctuation based on the  setting informations provided in Table1, dynamic values might follow a multinomial distribution. ( Lines 403-406, Section 4.1.

We once again appreciate your first and second advice mainly dealing with domain-specific concerns. With your overall feedback , we seemingly have managed to provide concrete  information of the real world dataset for experimental research.